# Information-theoretic Task Selection for Meta-Reinforcement Learning

**Ricardo Luna Gutierrez**
School of Computing
University of Leeds
Leeds, UK
`scrlg@leeds.ac.uk`

**Matteo Leonetti**
School of Computing
University of Leeds
Leeds, UK
`M.Leonetti@leeds.ac.uk`

## Abstract

In Meta-Reinforcement Learning (meta-RL) an agent is trained on a set of tasks to prepare for and learn faster in new, unseen, but related tasks. The training tasks are usually hand-crafted to be representative of the expected distribution of test tasks and hence all used in training. We show that given a set of training tasks, learning can be both faster and more effective (leading to better performance in the test tasks), if the training tasks are appropriately selected. We propose a task selection algorithm, Information-Theoretic Task Selection (ITTS), based on information theory, which optimizes the set of tasks used for training in meta-RL, irrespectively of how they are generated. The algorithm establishes which training tasks are both sufficiently relevant for the test tasks, and different enough from one another. We reproduce different meta-RL experiments from the literature and show that ITTS improves the final performance in all of them.

## 1 Introduction

One of the main challenges presented in Reinforcement Learning (RL) is to be able to perform optimally across a range of tasks, especially in the real world, where experience is expensive, and a poor initial behaviour may result in significant costs. The trained agent would ideally be able to adapt to slight variations in the dynamics of the environment without retraining, and show close to optimal performance from the beginning of the new task.

Meta-learning in RL has been gaining popularity as a methodology to face such challenges, and train agents requiring minimal online adjustments when a variation of a previously seen task is presented [7; 30; 8; 22]. In meta-learning, an agent is trained on a wide set of tasks so that it can learn their common features, and transfer knowledge to unseen tasks that share some properties with the training set. Meta-RL has also had promising success in real-world scenarios, when adapting to new tasks [20; 2; 23].

Training tasks are designed with the intention of being representative of a family of test tasks, or a skill the agent is expected to learn. A common framework consists in modeling the range of tasks the agent may encounter as a distribution over all possible tasks. Existing meta-RL methods use dense coverage of task distributions, generating a very large set of training tasks used to meta-learn a policy that can quickly adapt to new, unseen, tasks. However, dense sampling of tasks is highly computationally expensive, and in some cases infeasible. Standard meta-RL methods have not considered these scenarios where a limited number of training tasks is available. Furthermore, they have so far not considered that the training tasks may not be equally informative, beneficial, or promoting generalization. In this paper, we show that when a limited set of training tasks is available for training, not all tasks are necessarily beneficial, and selecting a subset of training tasks may lead to a better performance in the test tasks.

We introduce an Information-Theoretic Task Selection (ITTS) algorithm, that filters the set of training tasks identifying a subset of tasks that are both *different* enough from one another, and *relevant* to tasks sampled from the target distribution. The method is independent of the meta-learning algorithm used. The outcome is a smaller training set, which can be learned more quickly and results in better performance than the original set.

Task selection is performed before meta-learning and in conjunction with an existing meta-learning algorithm. We identified five domains in the literature that have been used to assess existing meta-RL algorithms, and evaluated ITTS in the same settings. The results show that task-selection improves the performance of two meta-RL algorithms (RL$^2$ [7] and MAML[8]) in all domains. We also introduce a sixth domain as an example of a real-world application on device control for micro grids, and use it to validate our approach in a realistic setting.

## 2    Background and Notation

We consider the classical RL setting, where a task is represented as a Markov Decision Process (MDP) $m = \langle S, A, R, P, \gamma, \mu \rangle$, where $S$ is the set of states, $A$ is the set of actions, $R \subseteq \mathbb{R}$ is the set of possible rewards, $P(s', r|s, a)$ is a joint probability distribution over next state and reward given the current state and action, $0 \leq \gamma \leq 1$ is the discount factor, and $\mu(s)$ is the initial state distribution. We assume that tasks are *episodic*, that is, the agent eventually reaches an *absorbing* state that can never be left, and from which the agent only obtains rewards of $0$. In our framework states and actions can be continuous, but we will use the discrete notation for simplicity. The behaviour of the agent is represented by a policy $\pi(a|s)$ returning the probability of taking action $a$ in state $s$. An MDP $m$ and a policy $\pi$ induce an on-policy distribution $d_\pi^m(s)$ as the fraction of time steps spent in $s$ during an episode. The expected number of visits to a state $s$ is $\zeta(s) = \mu(s) + \sum_{\bar{s} \in S} \eta(\bar{s}) + \sum_{a \in A} \pi(a|\bar{s})p(s|\bar{s}, a)$ for all $s \in S$. The previous system of equations can be solved for each state, and the on-policy distribution be computed as $d_\pi^m(s) = \frac{\zeta(s)}{\sum_{s' \in S} \zeta(s')}$. The goal of the agent is to compute an optimal policy $\pi^*$, which maximizes the expected return $G_t = \mathbb{E}[\sum_{i \geq t} \gamma^{i-t} R_{i+1}]$ from any state $S_t$ at time $t$. The value function $v_\pi(s) = \mathbb{E}_\pi[G_t]$ represents the expected return obtained by choosing actions according to policy $\pi$ starting from state $S_t = s$.

### 2.1    Meta-Learning

In meta-learning, the agent trains on a set of tasks to acquire knowledge that allows it to adapt quickly to a new task. Given a possibly infinite set of tasks $\mathcal{M}$ and distribution over them $p(\mathcal{M})$, the agent has access to $T$ training tasks $\mathcal{T} = \{m_i\}_{i=1}^T$. At meta-test time, a new task $m_j \sim p(\mathcal{M})$, such that $m_j \notin \mathcal{T}$ is extracted from $p(\mathcal{M})$, and the agent is expected to learn the optimal policy of the new task with a small number of samples. Since our method relies on policy transfer, we require that a policy learned in any of the tasks is also defined in every state of all the other tasks in $\mathcal{M}$. This is a common requirement in meta-RL, and it can be achieved either assuming that all tasks in $\mathcal{M}$ have the same state and action spaces, or at least that they have the same dimensions.

At an architectural level, meta-learning is usually described as a combination of two learning systems, a lower-level system which is responsible for adapting to new tasks and has relatively fast learning time, and a higher-level system that slowly learns across the set of training tasks to improve the lower-level system [30]. Meta-learning in RL can be divided into two categories [22]: context-based methods and gradient-base methods. In context-based methods a model is trained to use history as a form of task-specific context [7; 30; 19]. These kinds of models make use of internal memory, such as recurrent neural networks, to enable the policy to adapt based on the dynamics, rewards and actions of the given task. In contrast, gradient-based methods learn from experience using meta-learned loss functions, hyperparameters or policy gradients [8; 27; 31; 32; 13; 28]. We selected one of the most popular methods in each category, namely RL$^2$ [7] and MAML[8], for the experimental evaluation.

## 3    Related Work

Meta learning is a growing research field quickly gaining popularity in the past few years. Many successful reinforcement learning applications, such as navigation tasks [8; 7; 30; 19], classic control tasks [17; 24; 31; 28], and locomotion tasks [8; 22; 11], have demonstrated the potential of using

meta-learning in RL. Despite the strong results obtained, these methods do not take into account cases where a small set of training tasks is available, or an optimal selection of tasks, and thus they do not measure how the presence of each training task affects the final performance of the agent.

Source task selection has been studied in the scope of transfer learning for RL before. For instance, measurement of similarity between MDPs [1; 14; 16; 26] have been employed to increase positive transfer between a single source and test task. Similarly, task clustering [4; 18] has been explored, selecting and creating an optimal set of tasks that can improve the performance of the agent in a final task. These approaches look for similarity between tasks, and select source tasks for transfer or clustering based solely on their proximity. However, it has been shown in multi-task learning that training on tasks that are very similar could lead to overfitting [34].

Optimal task selection has also been considered in the context of Curriculum Learning [25; 21; 29; 9], in which tasks are selected and ordered to build an optimal task sequence to train for a given set of test tasks. These approaches select and sort tasks to be of increasing complexity, so as to identify the best sequence that maximizes sample efficiency and/or final performance in the test tasks. However, the curriculum learning training is sequential and tasks are used one at a time, transferring the policy between them. This training process is different from meta-learning, where the agent is trained on all the tasks at the same time. Furthermore, these methods do not take overfitting into account, as generalization to a potentially large number of unseen test tasks is not the main focus for a curriculum.

Unsupervised task discovery has been explored in the scope of Meta-RL [15; 12]. In these approaches, new training samples are generated in the meta-training process by introducing variations of a reward function to the MDPs used for training. These methods act on the meta-training process and do not take into account the selection of existing tasks.

In this work, we propose a source task selection method tailored for meta-learning, agnostic of the methods used for task generation and meta training.

## 4  Source Task Selection for Meta-Learning

Our method is executed in conjunction with an existing meta-RL algorithm, and therefore falls into the meta-RL framework introduced in Section 2.1. We make two further assumptions: that training tasks $\mathcal{T}$ can be learned to convergence, and therefore their optimal policies $\{\pi_i^*\}_{i=1}^T$ are available; that the state space of training tasks can be sampled, which will allow us to estimate the difference and relevance as introduced in this section.

ITTS takes as input two sets of tasks sampled from $p(\mathcal{M})$. The first set of $T$ tasks is the training set $\mathcal{T}$ common to all meta-learning algorithms. The second set, $\mathcal{F} = \{m_j\}_{j=1}^K$ of $K$ tasks, such that $m_j \sim p(\mathcal{M})$ and $\mathcal{T} \cap \mathcal{F} = \emptyset$ is the *validation* set. The intuition behind ITTS is that the most useful subset of $\mathcal{T}$ contains tasks that are both *different* enough from one another, and *relevant* for the validation tasks. In the rest of this section we translate this intuition into a heuristic algorithm.

We take an information-theoretic perspective to measure the difference and relevance of training tasks, based on the policies that the agent learns for them. We define the difference between two training tasks $m_1$ and $m_2$ as the average KL divergence of the respective policies over the states of the validation tasks:

$$\delta(m_1, m_2) := \frac{1}{K} \sum_{m_j \in \mathcal{F}} \frac{1}{|S_j|} \sum_{s \in S_j} \sum_{a \in A} \pi_{m_1}^*(a|s) \, log \frac{\pi_{m_1}^*(a|s)}{\pi_{m_2}^*(a|s)}. \tag{1}$$

To define the relevance of a task $m_1$ to a task $m_2$, we consider the optimal policy of $m_1$, $\pi_{m_1}^*$, and its transfer to $m_2$. The policy obtained after $l$ episodes of learning in $m_2$ starting from $\pi_{m_1}^*$ will be denoted as $\pi_{m_1, m_2}^l$. We define relevance as the expected difference in entropy of the policies before and after learning, over the states of the validation tasks, with respect to the on-policy distribution before learning:

$$\rho_l(m_1, m_2) := \mathbb{E}_{s \sim d_{\pi_{m_1}^*}^{m_2}, \pi_{m_1, m_2}^l} \left[ H(\pi_{m_1, m_2}^l(a|s)) - H(\pi_{m_1}^*(a|s)) \right]. \tag{2}$$

The ITTS algorithm is shown in Algorithm 1. Before execution, $n$ states are sampled uniformly from the tasks in $\mathcal{F}$ and stored in a set of validation states $S_v$. This sample of states will be used to estimate

$\delta$ from Equation 1, in place of the sum over all states for all validation tasks. The set of training tasks $\mathcal{T}$, validation tasks $\mathcal{F}$, and sample states $S_v$ are given in input to the ITTS algorithm.

---

**Algorithm 1** Information-Theoretic Task Selection

---

1: **Input:** $\mathcal{T}$ all available task and $\mathcal{F}$ validation tasks, $S_v$ sample states, $\epsilon$ difference threshold, $i$ iterations of initial policy , $l$ learning episodes.
2: **Output:** $\mathcal{C}$ optimal meta training source tasks
3: $\mathcal{C} \leftarrow \{\}$
4: **for** $t$ **in** $\mathcal{T}$ **do**
5:     different $\leftarrow true$
6:     **for** $c$ **in** $\mathcal{C}$ **do**
7:         $\delta_c \leftarrow \frac{1}{n}\sum_{s \in S_v} D_{KL}(\pi_t^*(a|s), \pi_c^*(a|s)) \geq \epsilon$
8:         different $\leftarrow$ different $\wedge\ \delta_c$
9:     **end for**
10:     relevant $\leftarrow$ RelevanceEvaluation$(\pi_t, \mathcal{F}, i, l)$
11:     **if** different $\wedge$ relevant **then**
12:         C $\leftarrow$ C $\cup \{t\}$
13:     **end if**
14: **end for**

---

The subset of selected training tasks $\mathcal{C}$ is initialized with the empty set (line 3). Each task $t \in \mathcal{T}$ is then evaluated for difference from the tasks in $\mathcal{C}$ and relevance with respect to the validation tasks. The algorithm computes an estimate of $\delta(t, c)$ for tasks $t \in \mathcal{T}$ and $c \in C$ and tests whether it is greater than or equal to a parameter $\epsilon$ (line 7).

---

**Algorithm 2** RelevanceEvaluation

---

1: **Input:** $\pi_t$, $\mathcal{F}$, $i$ learning epochs in validation task, $l$ learning episodes
2: **Output:** $isRelevant$
3: $isRelevant \leftarrow False$
4: **for** $f$ **in** $\mathcal{F}$ **do**
5:     $\eta_b \leftarrow 0, \eta_a \leftarrow 0$
6:     **for** 1 to $i$ **do**
7:         $S_e \leftarrow \text{execute}(\pi_t^*, f)$
8:         $\pi_{t,f}^l \leftarrow train(\pi*_t, f, l)$
9:         $\eta_b \leftarrow \eta_b + \frac{1}{n}\sum_{s \in S_e} H_{\pi_t}(s)$         //Equation 4
10:         $\eta_a \leftarrow \eta_a + \frac{1}{n}\sum_{s \in S_e} H_{\pi_{t,f}^l}(s)$      //Equation 4
11:     **end for**
12:     $\hat{\rho}_l(t, f) \leftarrow \eta_a - \eta_b$
13:     **if** $\hat{\rho}_l(t, f) \leq 0$ **then**
14:         **return** true
15:     **end if**
16: **end for**
17: **return** false

---

ITTS then proceeds to check for relevance (line 10), as shown in Algorithm 2. The agent executes the optimal policy of a training task $t$ in a validation task $f \in \mathcal{F}$ for a number of episodes, generating a set of $n$ traversed states, which are stored in $S_e$ (line 7). This set is a sample of the states according to the on-policy distribution, used in estimating the expectation in Equation 2. Then the agent learns for $l$ episodes, starting from the transferred policy $\pi_t^*$, resulting in a learned policy $\pi_{t,f}^l$ (line 8). The entropy of both the transferred policy and the learned policy is evaluated on the set of states $S_e$ (line 9 and 10) by:

$$H_\pi(s) = -\sum_a \pi(a|s) log \pi(a|s).$$

The learning process is repeated $i$ times, sampling $i$ policies which are used to estimate the expectation with respect to the learned policy in Equation 2. If learning produced on average a policy of lower

entropy (therefore an information gain) we consider the training task that provided the transfer policy as *relevant* (line 14 in Algorithm 1).

If a task $t \in \mathcal{T}$ is different from all the currently selected tasks in $\mathcal{C}$ and relevant for at least a validation task in $\mathcal{F}$ it is added to $\mathcal{C}$ (line 12 in Algorithm 2).

# 5 Experimental Evaluation

The main aim of this evaluation is twofold: to demonstrate that task selection is indeed beneficial for meta-RL, and show that applying ITTS to existing meta-RL algorithms consistently results in better performance on test tasks. We also analyze the effect of the main algorithm parameter, $\epsilon$, on the results, and perform an ablation study to show that both difference and relevance contribute to the performance of ITTS.

As mentioned in Section 2.1 we used RL$^2$ [7] and MAML[8] to execute in conjunction with ITTS, each one being a popular algorithm from the category of context-based and gradient-based method respectively. However, instead of following the standard meta-training process in which an agent have continuous access to a dense distribution of training tasks, a fixed set of tasks is sampled for training. We surveyed the literature to identify domains used to demonstrate meta-RL algorithms, with the following two characteristics: available source code, and programmatic access to the task distribution $p(\mathcal{M})$ to generate further tasks. We identified five such domains: CartPole [17; 31; 28], MiniGrid [6], Locomotion(Cheetah) [8], Locomotion(Ant) [8], and Krazy World [27]. CartPole and MiniGrid are less computationally demanding, and have been used for the parameter and ablation studies. The other domains are more complex control problems and have been used, in addition to the previous ones, to evaluate the effectiveness of ITTS in the same setting as either RL$^2$ or MAML[8]. We also introduce a sixth domain, MGEnv, as a representative of a realistic application in micro-grid control.

In the rest of this section, we first introduce the domains, then evaluate the effect of the threshold $\epsilon$, show the results of the ablation study, and lastly we apply ITTS to RL$^2$ and MAML in their respective domains.

## 5.1 Domains

In this section, we introduce the experimental domains, and the process used to generate the tasks, common to train, validation, and test sets. We limited the number of training tasks in each domain so that the generation and training until convergence repeated for $5$ times would not exceed 72 hours of computation on an 8-core machine at 1.8GHz and 32GB of RAM. As a consequence, simple domains like CartPole have more training tasks than more complex domains, like MGEnv. In every domain we used $K = 5$ validation tasks.

### 5.1.1 CartPole

CartPole, from OpenAI gym [3], is a classic control task in which a pole is attached by an unactuated joint to a cart which moves along a track. The goal is to prevent the pole from falling over. The cart is controlled by applying a positive or negative force, and a reward of +1 is obtained for each time step the pole remains upright.

For this domain, 60 different training tasks were created by sampling the parameters of the environment from a uniform distribution. The parameters are: the length and the mass of the pole, the mass of the cart, the intensity of gravity, the quantity of force applied to the cart in an action, and the degrees from the vertical position in which the pole is considered as fallen.

### 5.1.2 MiniGrid

MiniGrid is an open-source grid world package proposed as an RL benchmark [5]. In the grid world used for our experiments, the agent navigates a maze with the purpose of reaching a goal state. The agent receives a reward between 0 and 1 when it reaches the objective, proportional to the number of time steps taken to reach it. For this domain, 34 training tasks were created by randomly changing the shape of the maze, the initial position, and the goal.

### 5.1.3 Locomotion

The locomotion domains are borrowed from the presentation of MAML[8]. In these domains two simulated robots, a planar cheetah and a 3D quadruped ("ant"), are required to run at a particular velocity. The reward is the negative absolute value between the current velocity of the robot and the goal velocity. This goal velocity is chosen uniformly at random between 0.0 and 2.0 for the cheetah and 0.0 and 3.0 for the ant. 40 training tasks were created for each domain. The goal velocity on test tasks was chosen randomly and uniformly.

### 5.1.4 Krazy World

Krazy World Stadie et al. [27] is a grid world domain proposed to test adaptation of meta-RL agents. In this domain the agent explores the environment looking for goal squares which provide a reward of +1 while evading obstacle squares which delay or kill the agent. A colour is assigned to each type of square for the agent to able to identify them. For each generated task, the position of the agent, the position of the goal and obstacle squares, and the colour assigned to them is randomly chosen. For this domain, 40 different tasks were used to build the training set.

### 5.1.5 MGEnv

The purpose of this domain is to validate our approach in a real-world scenario. We aim to demonstrate that our method is suitable for this kind of tasks, where the parameters of the environment can be highly variable and suboptimal performance is highly costly.

MGEnv is a simulated micro-grid domain created out of real data from the PecanStreet Inc. database. The domain includes historical data about energy consumption and solar energy generation of different buildings in the USA. Tasks in this domain are defined as a combination of three elements: the model of the electrical device to optimize, the user's schedule, specifying if the device must be run in given day, and the building data, containing the energy generation and consumption of the given building. The devices used for the experiments behave as a time-shifting loads, meaning that, once started, they run for several time steps and cannot be interrupted before the end of their routine. The goal of the agent is to find the best time slot to run the given device, optimizing direct use of the generated energy, while following the user's device schedule. The agent receives a positive reward when it uses energy directly from the building generation and a negative reward when consuming energy from external sources. The reward amount depends on the quantity of energy used from both sources. A reward of $-200$ is obtained if the device is not run accordingly to the user schedule. The simulator replays real data of energy generation and consumption of the building (other than of the simulated controlled device), and is therefore as close as possible to running the device in the real building at that time.

For this domain 24 tasks were used to build the set of training tasks. These tasks represented 24 buildings with different energy consumption, energy generation, schedule, and devices' energy consumption and running time. Test tasks were selected by choosing buildings that had an energy generation and consumption levels at most 80% similar to the ones presented in the validation set, while the devices' properties were different for all the tasks in both sets.

### 5.2 Results

A full experimental run proceeds as follows: tasks are selected according to ITTS, the meta-learning algorithm is run to obtain a meta-policy, and the policy is evaluated in the test tasks. $RL^2$ [7; 30] was used as the meta-RL algorithm in Krazy World and MGEnv, in addition to the parameter and ablation studies. MAML was used in Ant and Cheetah. In both cases, we aimed at reproducing the results of the papers where the domain is presented. For all domains we show the average performance over all the runs. This accounts for slight discrepancies with respect to some of the original papers when they show the best model, rather than the average[1]. In all the plots the error bars are 95% confidence intervals.

## 5.3 Parameter Evaluation

We start by studying the effect of the main threshold parameter, $\epsilon$, of the algorithm in the two least computationally expensive domains, which allows us to repeat the learning process 10 times over 5 test tasks, with a total of 50 different test tasks. The threshold determines when a task is considered different enough from another task, that is, their difference measured as in Equation 1 is greater than or equal to $\epsilon$. The results are shown in Figure 1a and 1b for CartPole and MiniGrid respectively. The plots show that the optimal value is domain dependent, but rather easy to determine, since the return in the final tasks with respect to the parameter $\epsilon$ is convex. A parameter of $\epsilon = 0$ corresponds to ignoring task difference, and considering only task relevance. The optimal parameters established this way have been used in the rest of the experiments. For better comparison between domains, the $\epsilon$ values shown in the figures were normalized by the number of actions in each domain. Returns are also normalized between 0 and 1.

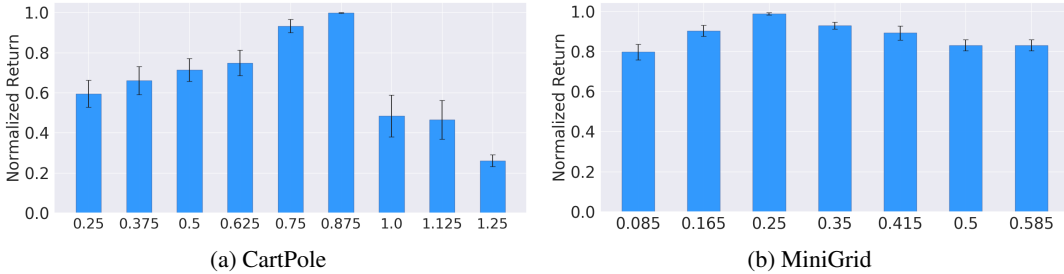

Figure 1: Results of parameter evaluation. Values shown on the x-axis represent the normalized values used for $\epsilon$ while the y-axis shows normalized returns

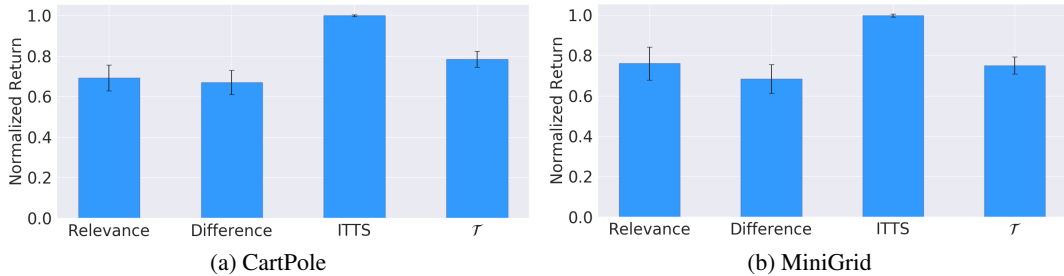

Figure 2: Ablation study. The plot shows average performance on test tasks of the agents trained using only relevance, only difference, and both (ITTS). "$\mathcal{T}$" is the performance obtained using all training tasks, without task selection.

## 5.4 Ablation Study

In this experiment we aim at establishing that both difference and relevance indeed contribute to the transfer. We evaluated the agent using only relevance, only difference, and both (ITTS). The results are shown in Figure 2a for CartPole and Figure 2b for MiniGrid. The results were obtained, similarly to the previous experiment, by evaluating the agent in 5 test tasks for 10 runs. In both domains, neither difference nor relevance alone can obtain the same performance than when used in combination (ITTS).

## 5.5 Transfer Results

Lastly, we evaluate the effect of ITTS on existing meta-RL algorithms on all six domains. In addition to the meta-RL algorithm with and without ITTS (using all the tasks in $\mathcal{T}$), we also consider as a baseline a random subset of the training tasks, and using the validation set $\mathcal{F}$ as the training set (without ITTS). The results are shown for CartPole in Figure 3, for MiniGrid in Figure 4, for Ant in Figure 5, for Cheetah in Figure 6, for KrazyWorld in Figure 7 and for MGEnv in Figure 8. The

agents were evaluated over 5 test tasks per run, with results averaged over 5 runs (25 different test tasks in total). The results of random selection are averaged over 4 random subsets of random size. In these plots as well, the shaded area is the 95% confidence interval. The results show the consistent performance improvement achieved by ITTS over the baselines. Interestingly, the set of training tasks is not always significantly better than the set of validation tasks (when used for training), despite the latter is much smaller. This also confirms that more tasks do not necessarily improve the final performance, and appropriate tasks must be selected.

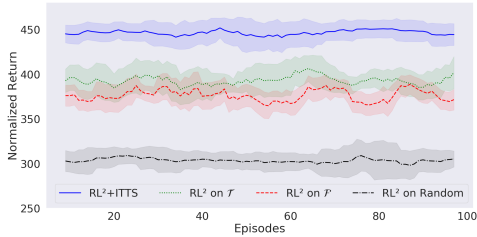

Figure 3: Results on CartPole domain.

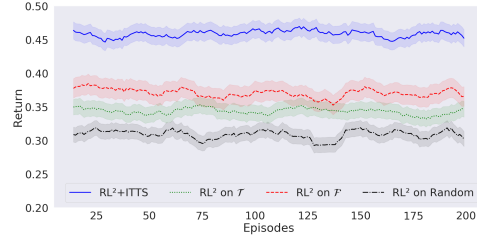

Figure 4: Results on MiniGrid domain.

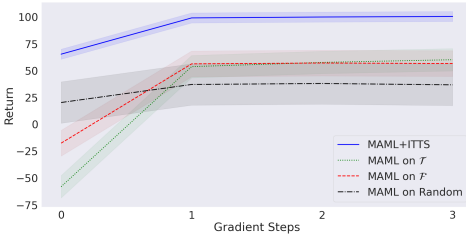

Figure 5: Results on Ant domain. 20 rollouts per gradient were used.

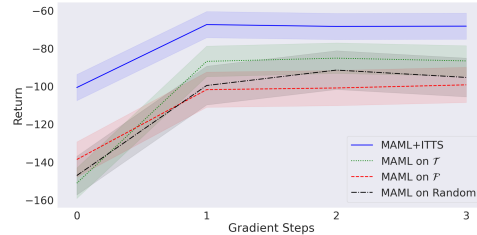

Figure 6: Results on Cheetah domain. 20 rollouts per gradient were used.

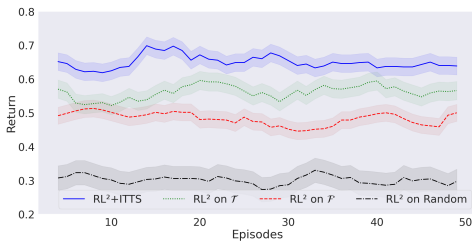

Figure 7: Results on Krazy World domain

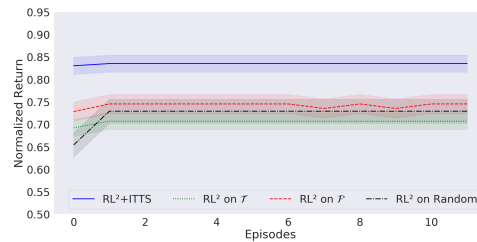

Figure 8: Results on MGEnv domain. Returns are normalized

# 6   Conclusion

We introduced an Information-Theoretic Task Selection algorithm for meta-RL, with the goal of improving the performance of an agent in unseen test tasks. We experimentally showed that an agent trained using a subset of tasks selected by our algorithm outperforms agents that were trained using all the available training tasks or random subsets of these tasks. We also showed how both *difference* and *relevance* are important in ITTS for boosting the performance of the agent.

The presented results unequivocally demonstrate the potential of task selection in meta-RL, which has been so far overlooked. The heuristic nature of our algorithm raises the question of a theoretical understanding of the role of tasks on generalization in meta-RL: a promising area for future work.

# 7 Broader Impact

This work is tied to current meta-RL algorithms, and thus its social impact is directly related to meta-RL itself. In meta-RL, the final goal is to train general agents that can perform in many different tasks with low adaptation time required. Recent work has shown that this goal is still a long way away [33], but progress in the field could have an important presence in real-world tasks in the near future. Increasingly general agents could lead to an acceleration in the deployment and training of smart systems, saving time and energy, and assisting human decision makers and workers in an evergrowing gamut of tasks. General considerations on desirable safety of AI systems apply to meta-RL as well.

## Acknowledgement

This work has taken place in the Sensible Robots Research Group at the University of Leeds, which is partially supported by EPSRC (EP/R031193/1, EP/S005056/1).

## Footnotes

[1]All the parameters and implementation details for every experiment are available in the supplementary material, as well as the source code. For training individual tasks and meta-RL agents, garage [10] was used.

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
