[Supplementary Material · SuplementaryMaterial.pdf]

# A    Additional Experiment Details

In this section, we provide details of the experimental set-up.

## A.1    Individual Tasks

To get the optimal policy of the individual tasks in CartPole, MiniGrid, Krazy World and MGEnv, PPO was used as training algorithm while TRPO was used for Ant and Cheetah. For all the experiments we used Multilayer percepton with two hidden layers of 32 units each and learning rate of 0.001.

## A.2    Meta-Training

For the $RL^2$ agents a Gated Recurrent Unit and two Dense layers of 64 units were trained with PPO. The clip-range and learning rate were set to $0.2$ and $0.001$ respectively. For the MAML agents a Multilayer percepton with two hidden layers of 64 units each was trained using TRPO. As in the original MAML implementation the inner learning rate was set to 0.1.

## A.3    Task Selection

The parameters used to apply ITTS in all the domains were; number of total learning episodes $l = 100$ for CartPole, MiniGrid, Krazy World and MGEnv and $l = 200$ for Ant and Cheetah, number of state samples $|S_v| = 500$ (100 per validation task), number of learning epocs in validation tasks $i = 5$.

# B    Ablation Study

We provide exact numbers of the results obtained by the ablation study for better clarity.

|  | CartPole | MiniGrid |
|---|---|---|
| Relevance | 0.693 | 0.760 |
| Difference | 0.670 | 0.683 |
| ITTS | 0.997 | 0.995 |
| $\mathcal{T}$ | 0.784 | 0.750 |

| CartPole | | MiniGrid | |
|---|---|---|---|
| $\epsilon$ | Return | $\epsilon$ | Return |
| 0.25 | 0.594 | 0.085 | 0.798 |
| 0.375 | 0.659 | 0.165 | 0.903 |
| 0.5 | 0.713 | 0.25 | 0.988 |
| 0.625 | 0.748 | 0.35 | 0.929 |
| 0.75 | 0.932 | 0.415 | 0.892 |
| 0.875 | 0.998 | 0.5 | 0.831 |
| 1.0 | 0.483 | 0.585 | 0.831 |
| 1.125 | 0.464 | | |
| 1.25 | 0.260 | | |

# C    Tasks Parameters

We show isomaps of the parameters in training, validation, test and selected tasks of CartPole, MiniGrid and KrazyWorld. For CartPole we use the numeric parameters of the tasks as input of the isomap algorithm, for MiniGrid and KrazyWorld we use an image of the initial state. Moreover, we plot the velocities used for Ant and Half-Cheetah.

Figure 1: Parameters of CartPole domain.

Figure 2: Parameters of MiniGrid domain.

Figure 3: Parameters of Half-Cheetah domain.

Figure 4: Parameters of Ant domain.

Figure 5: Parameters of KrazyWorld domain.