[Reviews · NeurIPS 2020]

Review 1

Summary and Contributions: [UPDATE] I have read the rebuttal, and I still believe the authors should work on experiment description clarity. I do not dispute that this paper has committed the common sin of saying "We assume the standard meta-RL framework" and moving on. However, I believe three points are in favour of this paper: - The authors' response seems to indicate that the reviewers' message has been heard and more details are going to be included; I would actually prefer they did not clutter the main paper with these details because ... - The meta-RL methodology for these tasks is very well known and "standard" so if they made changes, it's likely that they made the tasks harder, not easier. There are dozens, perhaps more, papers building on this methodology starting from 2017 onwards, many in top tier conferences, and a majority do not describe the tasks in detail in the main paper. - The authors put the gained space to good use, e.g. they include an ablation study and results on an unusually large bunch of domains, more than the average accepted paper. I would still like harder domains, but I can't disregard presented evidence (yet). As other reviewers have kindly pointed out, this paper's main point is not one of improved SOTA, but of a new angle on an active topic of research supported by reasonable evidence. In balance, I believe this should take precedence and I have decided to increase my score. However, I believe the authors *must* further highlight experimental differences to "standard" meta-RL and *must* motivate them! My score reflects that I trust in authors to do this! Suggestions: - Using a relatively small random sample of tasks rather than dense coverage of task distributions is non-standard. Line 171 in section 5.1 outlines this, but there is little motivation; this will simply not do and *must* be discussed at length. E.g. [1] clearly states that the onus of creating the meta-training task distribution which leads to meta-generalisation is on practitioners. However, such an endeavour is very difficult, and there is no manual with formal instructions. ITTS allows us (in principle) to automatically pick tasks which are useful from an "ad hoc" meta-training distribution. I believe sampling bias is inevitable with more complex tasks, since we can't easily predict how meta-learning changes when we add or subtract tasks to the distribution, and this process should be automated, hence ITTS and subsequent work should be useful. - Hence, it seems doable to argue for small random samples of tasks, since sample effects are probably even closer to real world experiments, which is why I welcomed the methodological change; also, it most likely makes meta-learning harder. However, it is on the authors to prove this and discuss it at length. - Motivate the paper in terms of meta-learning from a much more limited sample of tasks early on, rather than leaving this for section 5. - "MAML+ALL" should be labeled "MAML+FixedTasks" and it should be clarified in the paper that it is different MAML + dense tasks, which again is only minimally done in section 5. - List the tasks which were chosen, or better yet plot their parameters in figures. Some analysis to emphasize your numbers goes a long way. Your bar charts are ugly and don't need to take half a page to make the point, so no excuses please! Exact numbers should be in tables included in the appendix. - It has been mentioned that releasing code and checkpoints is doable with this framework, so please consider it. [1] Finn, Chelsea. Learning to Learn with Gradients. EECS Department, University of California, Berkeley. 2018. [OLD REVIEW] This paper investigates standard meta-reinforcement learning tasks and methods from a new angle: non-uniform task selection from the meta-training set. A novel heuristic algorithm is proposed: Information-Theoretic Task Selection (ITTS), which reduces the number of tasks that need to be considered in meta-training while empirically improving (meta-) generalization to meta-validation and meta-test sets of tasks. While the assumptions exploited in ITTS may not hold in future meta-RL task distributions, they do hold in a number of widely used benchmarks, and the paper provides convincing empirical evidence for two classes of meta-learning algorithms which form the basis for current state-of-the-art meta-RL approaches. While ITTS as presented may not generalise directly to more interesting task distributions, it is not hard to imagine approximations and efficient variations. All things considered, it is likely that future work will follow along the direction proposed here, despite the technical and theoretical challenges.

Strengths: The paper considers a somewhat idealized version of the meta-learning problem as used in the MAML and RL2 papers, yet still compatible with widely used benchmarks. It is assumed that meta-training tasks can be indexed and learned to convergence, such that an optimal policy per task index can be found and stored, and that state spaces of validation tasks can be easily enumerated, presumably weighted by state probabilities under optimal validation task policies. While these assumptions may seem excessive from an empirical point of view, they do not violate standard meta-learning assumptions, where full control over the meta-training distribution of tasks is allowed, as long as no meta-test information is accessed. I mention these details here to clarify that I find the methodology sound for a theoretical investigation, although not necessarily in the spirit of more gradious claims found in some meta-learning papers. I believe everyone working on meta-RL should read this paper, since results within go against common knowledge that dense sampling of task distributions leads to improved generalization. It may turn out that some of the assumptions may need approximations and the literal implementation of the heuristic algorithm may be inefficient for larger numbers of more complex task distributions with substantially more complex state spaces. However, ITTS may provide a means of designing better meta-training distributions. Since meta-training time currently scales at least linearly with numbers of tasks which need to be sampled, it is not impossible that ITTS may be useful even when its assumptions do not hold. This should lead to interesting follow-up work. Empirical validation is surprisingly clear. Virtuous interaction between training task dissimilarity and relevance to validation tasks is shown convincingly.

Weaknesses: This paper shares the weaknesses of other meta-RL papers and benchmarks. It is not yet clear how results on such simple "toy" tasks will, if ever, generalize to practically important task distributions. But this current limitation does and should not stop progress towards such seminal contributions.

Correctness: I believe the methodology is standard and compliant with the relevant literature.

Clarity: The paper is very clear and accessible. The proposed method is indeed largely agnostic to the underlying meta-RL algorithm, apart from adopting their standard assumptions, so lengthy descriptions of such methods is omitted without loss of clarity.

Relation to Prior Work: I found the related work section very up to date, despite most approaches not being directly relevant to the work. The section positions this work well within existing literature, nicely arguing for its complementary and largely orthogonal take on the problem of meta-training distributions.

Reproducibility: Yes

Additional Feedback:


Review 2

Summary and Contributions: The authors propose an algorithm that prunes a meta-reinforcement learning task set so that it better aligns with meta-validation (and hence, meta-test) tasks. Accepting a candidate task requires that it is both a) sufficiently distinct from other accepted meta-training tasks in terms of KL divergence between individual optimal policies, and b) relevant to at least one meta-validation task in terms of whether transfer learning of the candidate's optimal policy to the meta-validation task results in a reduction of policy entropy. Experiments on gridworld and simpler MuJoCo locomotion environments suggest that the pruning does have beneficial effect for downstream meta-RL.

Strengths: - Soundness of the claims: the empirical evaluation comprises two popular, distinct meta-RL algorithms and several experimental domains commonly used as benchmarks in the meta-learning literature. There are interesting ablation studies that probe the effect of the two component criteria of ITTS as well as the distinctiveness criterion threshold. - Signifiance and novelty: the problem the authors tackle is important and under-explored. The ITTS algorithm is novel and interesting. - Relevance: the work is relevant to those interested in meta-RL, automatic task design, and transfer learning.

Weaknesses: - Soundness of the claims: there is no theoretical justification for why the two criteria for task distinctiveness and relevance should work, and there are critical aspects of the experimental methodology that do not appear in the main text, supplement, or code. In particular, for each domain, what are the meta-training (un-pruned and pruned), meta-validation, and meta-test tasks? This information is crucial in helping to elucidate what tasks the algorithm prunes and why, as well as ascertaining the validity of the baseline comparisons.

Correctness: It is hard to assess the correctness of the empirical methodology, as each experiment comprise three main stages (i. learning optimal policies via RL for each of the meta-training and meta-validation tasks; ii. ITTS; iii. meta-RL on the pruned meta-training tasks) yet code is only provided for the second stage.

Clarity: There is an abundance of description of how the algorithm works, but a severe paucity of technical motivation/justification for the key design choices inherent in the computation of task difference and relevance (Eqs. 1 and 2).

Relation to Prior Work: This is done sufficiently.

Reproducibility: No

Additional Feedback: ---------------------------POST-RESPONSE COMMENTS--------------------------- - Thank you for your response. I have decided to raise my score to 4/4. Though the authors investigate an important, underappreciated problem, I remain convinced that the submission's execution, empirical analysis, and writing (motivation and discussion) are of insufficient quality; while the community may be inspired by the research questions the authors pose, upon closer inspection people will likely not be able to learn much from this submission as-is. I hope that the authors will find my comments helpful in improving this work towards being a significant contribution to the field. - During a spirited discussion period, I revisited the paper's methodology in detail, and discovered a few issues: a) For the locomotion experiments (ant and half-cheetah velocity-reaching), the authors start with a set of only 40 meta-training tasks and justify this with "As in the MAML paper, 40 training tasks were created for each domain." But this is not correct -- the MAML paper samples fresh tasks over the course of meta-training, instead of using a fixed set. The authors may have misread this line from the Appendix [8]: "in the locomotion problems, we used a meta batch size of 40 tasks". b) In Figs. 5 and 6, the baseline All+MAML results severely underperform. In MAML [8], we see returns of -60 for half-cheetah goal-velocity, and 100 for ant goal-velocity after 3 updates at meta-test time, compared to -90 and 50 for All+MAML. This is likely due to using a severely undersized training set from (a). c) In Figs. 5 and 6, the ITTS+MAML returns for ant match those of MAML [8] after 3 adaptation updates at meta-test time, but greatly surpass those of MAML after 0 updates. For half-cheetah, ITTS+MAML moderately underperforms MAML after 3 updates, but similar to ant, it greatly outperforms MAML at the meta-learned initialization. - From the above, here's my guess as to what's happening, at least for the locomotion experiments. We have a 1-d task space parameterized by the goal velocity. Because of (a), we have sampled a relatively small set of goal velocities (40) on a 1-d interval. Since the sample size is small, it is unlikely that the empirical distribution of samples looks uniform. ITTS prunes the 40 meta-training tasks using the difference metric, which removes some of the tasks that are closest together, making the resulting meta-training task distribution more uniform. Presumably, the meta-validation task velocities are selected to have uniform coverage of the interval; the text does not say that they are selected randomly. Hence, most or all of the meta-training tasks will be sufficiently relevant. The meta-test tasks are stated to be sampled from the uniform distribution over the interval, so, crucially, the ITTS-pruned closer-to-uniform meta-training task distribution is closer to the meta-test distribution than the unpruned set of 40 tasks. This also explains why ITTS+MAML does not outperform the original MAML results, which achieved a close-to-uniform meta-training task distribution by virtue of a large sample size. Finally, the difference in performance at 0 updates suggests that the finite task set results in meta-overfitting and/or a greater degree of memorization of the meta-training tasks. - All of the experiments use small meta-training sets. This is not surprising as ITTS requires a run of single-task RL for each meta-training and meta-validation task. But after considering the locomotion experiments, it may be that ITTS is mostly or only effective in the scenario of having small numbers of tasks in that it ameliorates distributional artifacts resulting from low sample sizes. This suggests that ITTS may be working for reasons unrelated to the motivation of the paper. - The salient points from the above conjecture are that i) it was indeed conjecture because of a lack of presentation of key results like the un-pruned and pruned meta-training task sets, and ii) this is the sort of dissection and analysis that is sorely missing from the paper. - Finally, I will consolidate from the reviewer discussion some key points of improvement for the submission: --- Empirical analysis of the task distributions: unpruned meta-training, pruned meta-training, meta-validation, and meta-test. Discussion on what an optimal meta-training set should look like (e.g. for a uniform meta-test distribution, we may want the meta-training set's empirical distribution to have maximum entropy). --- Motivation and justification (even on an intuitive level) when introducing ITTS, and discussion/interpretation of ITTS results e.g. in terms of reducing sampling bias, etc. in light of above results on the task distributions. --- Ensure that you follow the meta-RL methodology outlined in prior work, or clearly state (including priming the reader during the abstract/intro) why you choose to modify it with justification (e.g. when your conditions are more realistic/practical). --- Rename all+MAML to 40+MAML. Apply similar renaming to other domains where previous work used dense sampling of tasks over the course of meta-training. Include comparison to dense sampling (without ITTS) as an "oracle". --- Include code for full reproduction of experiments. --- Results on larger starting task sets for ITTS; just one or two experiments since this is expensive, and it's fine (even expected) if ITTS shows reduced or no improvement in this regime. (Optional but cool to have).


Review 3

Summary and Contributions: This paper proposes a metric and algorithm for selecting a curriculum of meta-train tasks, to be used with any meta-RL algorithm. It introduces the problem (meta-train task selection), gives some intuition for a solution (measuring relevance and difference between train tasks), and develops that intuition into a heuristic metric and simple task selection algorithm. It's claimed contributions are: * Formalization the task selection problem and empirical evidence it exists * State KL-divergence-based metric for measuring the relevance and difference between meta-train-tasks * A basic task selection algorithm which uses those metrics to increase performance on a set of small meta-RL tasks Response to Authors post-feedback ------- Thank you for your thorough response to my comments and the helpful explanation of both the limitations of the work, and for correcting my estimate of the time complexities. I agree that the limitations of your work are primarily driven by the limitations of existing meta-RL approaches and the problem setting, and have reconsidered my score. Thank you so much for bringing to the community's attention to this important question, which to my knowledge has been almost entirely neglected by recent meta/MTRL works so far.

Strengths: Soundness of claims ------ 3/5 This work is primarily empirical and heuristic in nature, so I will omit comments on theoretical grounding. I believe this works claims are sound and the empirical evaluation is well-considered and executed, if limited in scope. The primary claims are that task selection is important for performance (well-supported by citations and the experiments here), and that meta-RL performance can be significantly improved by carefully-choosing tasks. Secondary claims are their their selection method helps choose better tasks, leading to higher performance (also supported by the experiments) Significance, novelty, and relevance ---- 3/5 The algorithm appears novel, and while its implementation is not particularly efficient, it is generally-applicable and provides a good starting point for further study in this area. I believe the overall research problem discussed in this work does not get enough attention in the meta-RL community, and is therefore relevant and significant, even if the results are good-but-not-groundbreaking.

Weaknesses: The weaknesses of this work revolve around its limitations of applicability and the extent to which its empirical analysis might extend outside the toy domains from the empirical analysis. Simple benchmarks, with limited applicability of empirical analysis to future work, and potential for overfitting to simple benchmarks ----------- The authors show their method improving meta-RL test-time performance on a small panel of meta-RL challenges from the literature, however these are all very small "toy" problems which are well-known to be prone to overfitting. The simulated robotics tasks in particular just do not exhibit enough structural diversity to evaluate the applicability of this method to real robotics problems, or even more complex simulated robotics problems. Existing benchmarks exist ([1][2]) which offer much more diversity than the environments used by the authors. The authors even cite [2]. Using better benchmarks would provide the reader with more confidence in the extent to which these limited-scope empirical experiments might extend to her desired application domain. L157-159: " 157 We surveyed the literature to identify domains used to demonstrate meta-RL algorithms, 158 with the following two characteristics: available source code, and programmatic access to the task 159 distribution p(M) to generate further tasks. " [1] and [2] both provide access to source code and programmatic access to task distributions, but were not used. See for better benchmarks: [1] Hendersen, et al. 2017: https://arxiv.org/abs/1708.04352 [2] Yu, et al. 2019: https://arxiv.org/abs/1910.10897 Poor time complexity of the proposed algorithm --------------------- By my reading of Algorithms (1,2) suggest that the time/sample complexity of the proposed algorithm is somewhere in the neighborhood of O(T^2 * train()) or O(T^3 * train()), where T is the number of meta-test environments and train() is the amount of time (or number of environment samples) necessary to train an agent on a single instance of t \in T. Increasing the sample-complexity of already extremely sample-intensive meta-RL training algorithms by a factor of ^2 or ^3 is no small matter. It is important to ask, in real applications, what else could I do with an extra [(T - 1)*train()] or [(T^2 -1) * train()] samples to increase performance? Given that the performance increases in this empirical work are between 10 and 25% on toy problems, the answer is likely "a lot." For instance, what if I ran vanilla meta-RL training, but allowed the meta-RL algorithm to access T^2*train() meta-train samples (which is how many ITTS was allowed to use)? It's not clear whether ITTS is lowering sample complexity, or just hiding it somewhere else. Perhaps ITTS allows for better generalization in challenging adaptation scenarios which confound existing meta-RL algorithms (rather than the simple experiments here, which all have unimodal task distributions which are easy to perform inference on)? We have no way of knowing from this paper. Lack of discussion of limitations -------- The paper lacks a discussion of the method's limitations, which I think is a vital part of any methods work, particularly empirical ones.

Correctness: The empirical methodology appears correct and well-presented. I think that Figs 3-8 could benefit from an "oracle" line showing the best possible average performance on each environment.

Clarity: The paper is well-written and I find the presentation fairly lucid and clear. I appreciate the substantial work put into clearly motivating each section and the experiments. I recommend that the authors reconsider their naming scheme used for Figs 3-8. I had to read deep into the text to understand that "All" corresponds to conventional meta-RL training, "Random" is a variation on conventional meta-RL training, but with randomly-chosen subsets of tasks, and "Validation" is performance on the Validation set using conventional meta-RL. Perhaps you should separate the groupings into "Ours (ITTS)" and "Baseline (name-of-sampling-variation)" and include text for understanding these variations directly in the figure caption.

Relation to Prior Work: There is little prior work in this exact setting (the intersection of meta-RL + task selection), but there is a large body of work on task discovery and selection *outside* of meta-RL, and usually these works in RL and IL are relevant to works in meta-RL as well. * I believe the authors are remiss in not mentioning the body of work on the Options framework option discovery in hierarchical RL, e.g. see https://arxiv.org/abs/1611.07507 https://arxiv.org/abs/1609.05140v2 ..and their citations I think it would be appropriate to mention DADS (https://arxiv.org/abs/1907.01657) in the related work section when discussing unsupervised skill discovery.

Reproducibility: Yes

Additional Feedback: * Based on the supplemental, your experiments rely heavily on the garage reinforcement learning library, but you don't mention this in the paper text or cite the library. Please include the citation so that readers know which implementation you're using, and so the authors of the library are properly attributed.


Review 4

Summary and Contributions: The paper proposes an augmentation to existing meta-learning algorithms for selecting training tasks during meta-training. The key idea is that by selecting training tasks that are different from each other and relevant to the desired testing tasks, one can obtain a better training performance. To measure the difference between two training tasks, they used the KL divergence of the policies trained for the two tasks. To measure the relevance between the training task and a validation task, they used the difference between the policy entropy before and after adaptation to the validation task. The authors demonstrated the method on two meta-RL algorithms: MAML and RL^2 and showed that with the proposed method they can achieve better performance. The idea of selecting proper tasks for training meta-rl policies proposed in the paper is novel and interesting, which in my mind is the main contribution of the paper. The authors evaluated a specific criteria for selecting the training tasks and showed the effectiveness of it.

Strengths: One major issue with existing meta-rl algorithms is that they are usually limited to only work for the training tasks. By incorporating some knowledge about the tasks to be tested on (a validation set), the proposed method can potentially help generalizing existing algorithms to a wider set of problems.

Weaknesses: Though the proposed method seems effective and reasonable in general, more details regarding the problem setup can help better understand the performance of the algorithm and improve reproducibility of the work. In addition, comparing the difference between each pair of the training tasks seems expensive. Some discussions about the scalability of the method could be helpful.

Correctness: Yes

Clarity: Yes

Relation to Prior Work: Yes

Reproducibility: Yes

Additional Feedback: I have read the rebuttal and the other reviews. A key issue was raised during reviewer discussion that the presented setting (limited training task set) is different from the typical meta-learning setting. Although the current meta-learning setting can still be interesting, I feel more work are needed. I think there are two potential directions to make the work more solid: 1) evaluate and demonstrate that ITTS can outperform vanilla MAML with dense task sampling (scalability might be the bottleneck), and 2) focus the work on meta-learning with limited training tasks, demonstrate on problems where such setting is realistic, and perform more analysis on the proposed ITTS method (e.g. how sensitive is the algorithm to the number of training and validation sets, what tasks does ITTS selects and if any insights can be derived from them, etc.) In general the work looks at an interesting problem and has good potentials. ============================= Additional comments for the paper: 1) What is the intuition behind using entropy for measuring relevant, in contrast to using kl divergence between the two policies? 2) How are validation and testing tasks selected in the experiments? It is mentioned that the training tasks are sampled uniformly in certain range, but it’s not clear how the validation and testing tasks are selected. Are they from the same distribution that is far away from the training tasks? It would be nice to study how the performance varies when the distribution of training, validation, and testing tasks have different distances. 3) What is the baseline ‘all’? Does it contain the validation set? If not, why does having relevance or difference alone hurt the performance (Figure 2)? How would it perform if the meta-training is performed for the union of training and validation tasks? 4) In most tasks it seems that the learned policy before adaptation is already better than the baseline methods, which is a bit surprising. Are there any intuitions for that?

[Author Response · NeurIPS 2020]

**Paper ID 10791**
**Title: Information-Theoretic Task Selection for Meta-Reinforcement Learning**

We thank all the reviewers for their thoughtful feedback. Our response can be found below, organized by review.

**R1**
*"It is not yet clear how results on such simple "toy" tasks will, if ever, generalize to practically important task distributions. But this current limitation does and should not stop progress towards such seminal contributions."*

Thank you for the positive comments. We agree that scalability to more complex settings is challenging (more on this in response to Reviewer 3), but this is a challenge for all of meta-RL.

We introduce a method that identifies a clear gap in the literature, and that provides a first solution to the problem, which performs reliably well in a number of current meta-RL benchmarks. We don't expect it to be the last word on the subject, quite the opposite, we hope it will spur new research in its theoretical understanding, and that new meta-RL algorithms will incorporate a task selection component inspired by these results.

In this spirit, we agree with your view that *"It may turn out that some of the assumptions may need approximations and the literal implementation of the heuristic algorithm may be inefficient for larger numbers of more complex task distributions with substantially more complex state spaces. However, ITTS may provide a means of designing better meta-training distributions"*.

**R2**
*"there is no theoretical justification for why the two criteria for task distinctiveness and relevance should work"*

We state that the method is a heuristic, and acknowledge in the paper that further work is required on the theoretical side. However, many recent successes in machine learning have been driven by empirical results (most notably, deep learning), and we hope that the introduction of task selection into meta-RL can be one of those.

*" there are critical aspects of the experimental methodology that do not appear in the main text, supplement, or code. In particular, for each domain, what are the meta-training (un-pruned and pruned), meta-validation, and meta-test tasks? This information is crucial in helping to elucidate what tasks the algorithm prunes and why, as well as ascertaining the validity of the baseline comparisons."*

All the tasks, for training, validation, and test, have been generated according to the distribution provided by the environment. Every run had different tasks, amounting to hundreds of them across all domains. Looking at each one would be impossible, and we do not believe it to be crucial to the validity of the results. It is important that, as long as the tasks are extracted from the domain distribution, task selection improves meta-RL regardless of the particular tasks involved.

*"It is hard to assess the correctness of the empirical methodology, as each experiment comprise three main stages (i. learning optimal policies via RL for each of the meta-training and meta-validation tasks; ii. ITTS; iii. meta-RL on the pruned meta-training tasks) yet code is only provided for the second stage."*

The code for stage 1 and 3 is not ours, and in the readme file we provide links to the code repositories we used for the domains. For RL (stage 1) and meta-RL (stage 3) algorithms (TRPO, PPO, RL^2, and MAML) we used publicly available code by the respective authors, which can be freely downloaded.

**R3**

*"The authors show their method improving meta-RL test-time performance on a small panel of meta-RL challenges from the literature, however these are all very small "toy" problems which are well-known to be prone to overfitting."*

We did not mean to be limited to "toy" problems, but to use a range of tasks from the literature, whose results we could reproduce. It is a current limitation of state-of-the-art meta-RL algorithms that they have only be applied to such toy problems. Results with one domain could be an overfit, but we think that consistently good results over 6 domains can convince the reader that the benefits of ITTS are not the effect of the algorithm overfitting to any particular domain.

*"The simulated robotics tasks in particular just do not exhibit enough structural diversity to evaluate the applicability of this method to real robotics problems, or even more complex simulated robotics problems"*

This criticism is understandable, but should be directed at the original publications that proposed those domains. Again, we intended to reproduce and improve upon published results, to make clear that we did not design or use a particular domain because it exhibits the task-selection benefit, but rather that it is widespread in meta-RL, and already present in published domains.

*"Existing benchmarks exist ([1][2]) which offer much more diversity than the environments used by the authors. The authors even cite [2]. Using better benchmarks would provide the reader with more confidence in the extent to which these limited-scope empirical experiments might extend to her desired application domain."*

We did our best to use as many domains as possible with the required characteristics (access to code, programmatic change of parameters to create the task distribution) and representing a range of challenges in meta-RL, particularly focusing on sparse rewards (Krazy World and MiniGrid) and continuous control (the two locomotion tasks). We also introduced an application-inspired domain, to demonstrate a practical use of ITTS and meta-RL.

[1] was introduced in the multi-task learning context, and we could not find results with meta-RL algorithms on it that we could reproduce and improve upon. If we missed results on such a domain with RL^2, MAML, or other meta-RL algorithms that the reviewer is aware of, we would be grateful if they could point us to them.

[2] was a good candidate, and we did consider it. It did not make the final selection because it is at the edge of what meta-RL can achieve, and current algorithms struggle with this domain. Indeed the authors say "Our experiments show that current meta-RL methods in fact cannot yet generalize effectively to entirely new tasks and do not even learn the meta-training tasks effectively when meta-trained across multiple distinct tasks." We would have to use the subset of the domain in which current meta-RL algorithms perform well enough, attracting the same criticism the reviewer is making.

As reviewer 1 mentioned, *"This paper shares the weaknesses of other meta-RL papers and benchmarks. It is not yet clear how results on such simple "toy" tasks will, if ever, generalize to*

*practically important task distributions."* We agree with this view, in that we do share current limitations of meta-RL. We do not introduce ITTS to demonstrate that, only thanks to task selection, current meta-RL algorithms can generalize to much harder domains. We believe that more development will be required on of meta-RL algorithms. However, such future developments should take task selection into account, and ITTS provides a solution applicable to current domains, and a baseline for future work pushing the boundary of meta-RL applicability.

*"By my reading of Algorithms (1,2) suggest that the time/sample complexity of the proposed algorithm is somewhere in the neighborhood of $O(T^2 * train())$ or $O(T^3 * train())$, where T is the number of meta-test environments and train() is the amount of time (or number of environment samples) necessary to train an agent on a single instance of $t \in T$"*

The agent is trained once on every training task, with a cost of $O(T * train())$. This happens before Algorithm 1 is executed, so that it has access to $\pi^*_t$ for every $t \in T$. The computation of $\delta_c$ is $O(T^2)$ since the difference is computed $T*C$ times, and C is a subset of T. This computation, however, is the KL divergence of the policies on the samples of the validation tasks, and does not depend on the training time of the tasks. It is proportional to the number of states used for the estimate of the KL divergence, which is up to the user, with a more accurate estimate requiring more samples. The last step is the computation of relevance which requires $O(T*F*l)$ steps, where $|F| << |T|$ is the set of validation tasks. The parameter l is the number of episodes for which the transfer policy is learned, to give an estimate of the speed-up that transferring from that training task gives. This value is, again, much lower than the number of episodes required to converge to the optimal policy (the value of all parameters used in the experiments is in the supplementary material).

The complexity is dominated by the initial $O(T*train())$ computation, that is, by learning the optimal policy for all training tasks. Even in complex tasks, where the estimate of the KL divergence may require a large number of samples, learning the optimal policy for those tasks will still be significantly more expensive. For complex training tasks this is a significant limitation, but we believe that it can be accelerated, for instance not learning each training task from scratch as we did in this paper, but using transfer learning, or curriculum learning. However, we wanted these results to be independent of specific optimizations which may be required on complex domains.

To show the immediate applicability of the method, we also introduce the MGEnv domain, which is simple enough to allow the application of ITTS as presented (without further optimizations, for instance on learning the optimal policies) but is a realistic application scenario, with real data of energy generation and consumption. Admittedly, learning policies for micro grid control is less complex than many robotics tasks, but still economically significant. The simulation of the learned policies uses data of real buildings, and is as close as possible to actually running the simulated device in that building, at that time (data are from January 2016 to December 2018; the PecanStreet database is publicly available).

We would also like to thank the reviewer for the suggestions on clarifying figure naming, adding an "oracle" line, adding option papers to the related work section, and for catching the missing citation of the garage library. The last point, in particular, was indeed an oversight on our part, for which we apologize. We'll incorporate the suggestions in the next version of the paper.

**R4**
*"Though the proposed method seems effective and reasonable in general, more details regarding the problem setup can help better understand the performance of the algorithm and improve reproducibility of the work."*

We would be grateful if the reviewer could elaborate on what details they feel are missing, so that we can improve the paper. We answer the questions in Section 8 of the review below, hoping that this provides all the required clarification.

*"In addition, comparing the difference between each pair of the training tasks seems expensive. Some discussions about the scalability of the method could be helpful."*

Please see our response to Reviewer 3. The main computation bottleneck is learning the optimal policies for all training tasks. This limitation is discussed in the paper, and is the main barrier to the applicability of the method. Whether or not all training tasks can be learned in a reasonable amount of time, from the user's perspective, is indeed domain-dependent.

*"1) What is the intuition behind using entropy for measuring relevant, in contrast to using kl divergence between the two policies?"*

We did try the KL divergence for that comparison as well, but it does not lead to results as good as entropy. The problem is that the learned policy may differ substantially from the transfer policy, while not being the result of any useful learning. For instance, if the transfer policy is not useful in the target task, this may lead to catastrophic forgetting, and the learned policy degenerating to close to random exploration. This effectively corresponds to erasing all transferred knowledge and starting from scratch, which is not desirable, but gives a high KL divergence. However, if learning leads to a decrease in entropy, the learned policy is "sharper" than the transfer one in making decisions, which indicates progress towards an (at least locally) optimal policy. A reduction in entropy is a good indication of transfer being beneficial, corresponding to an information gain.

*"2) How are validation and testing tasks selected in the experiments? It is mentioned that the training tasks are sampled uniformly in certain range, but it's not clear how the validation and testing tasks are selected. Are they from the same distribution that is far away from the training tasks? It would be nice to study how the performance varies when the distribution of training, validation, and testing tasks have different distances."*

We assume the standard meta-RL framework, with only one task distribution. All tasks are generated by sampling from this distribution. For each domain we specify which parameters vary among the tasks, giving raise to the task distribution for that domain.

*"3) What is the baseline 'all'? Does it contain the validation set? If not, why does having relevance or difference alone hurt the performance (Figure 2)? How would it perform if the meta-training is performed for the union of training and validation tasks?"*

The baseline "all" is all the training tasks T. This name confused another reviewer, so it is clear that we need a better label for it. It does not contain the validation tasks. Training and validation tasks are extracted from the same distribution, so the union of both is just a larger training set from that distribution. We already show that larger training sets are not necessarily better. The use of the validation set for training demonstrates that those tasks are not "special" or more informative.

*"4) In most tasks it seems that the learned policy before adaptation is already better than the baseline methods, which is a bit surprising. Are there any intuitions for that?"*

RL^2 does not perform online learning, but the first few episodes are used to fill in the memory of the recurrent neural network with the new context. The network has been entirely learned during meta-learning, so it is perhaps less surprising. For what concerns MAML there is indeed learning in

the test tasks, and the policy learned with ITTS does have a better starting value. We do not have any particular intuition about why this is the case. We can only note in the results that the learned policy does indeed generalize better to new tasks from the first episode.

[Meta-Review · NeurIPS 2020]

This paper was quite controversial among the four reviewers, leading to more than 10 pages of discussion (longer than the paper itself!) In the end, two reviewers were advocating for acceptance (R1, R3), one was advocating for rejection (R2), and one was leaning towards rejection (R4). [Note that R4 did not update their score/review, but participated in the discussion] The main pros of the paper are: + The paper studies a new direction of selecting tasks in meta-RL. This is a direction that hasn't been studied before, and will likely become quite relevant in settings where the task distribution is quite heterogeneous + The experimental results suggest that the algorithm performs very well on a large number of simple domains, when combined with MAML and RL^2. + The experiments also include an ablation study. + Time complexity is not an issue; the reviewers appreciated the author response here. These are the main reasons that R1 and R3 were advocating for acceptance. I agree that these are strong points, and make me want to accept the paper. However, there are also a number of weaknesses. The overall execution of the paper is lacking: - There is no theoretical motivation, making it difficult to understand the intuition behind why this method should work. In principle, this should not be a deal-breaker on its own. However, see the next point. - There is limited empirical analysis, which makes it difficult to understand *why* the algorithm works well. - The combination of the two points means that it may be difficult for the reader to draw useful lessons or take-aways from method or experiments in the paper. - The quality of the writing could be significantly improved. - The experiments section doesn't sufficiently describe the experimental set-up See the updated reviews for details on each of these points. The reviewers came up with very specific recommendations for how to improve the paper based on these points. Because the reviewers were unable to come to a consensus, I also took a look through the paper. I have some concerns about the quality of the writing in the methods section and left some detailed comments below. In the balance, I think that the ideas in this paper are worthy of publication, but I urge the authors to improve the writing and analysis in the paper based on the reviewer's detailed feedback and the feedback below. ------------------------------ Feedback on the Writing ------------------------------ Methods section - \mathcal{F} is referred to as both the validation tasks and the test tasks. Which is it? In most places, it is referred to as the validation tasks. In this case, are there a separate set of test tasks that are used for evaluation? If these tasks are actually used in the evaluation, then they should be called the test tasks. - On line 127, it says that n states are sampled uniformly from the tasks in F. What policy collects these states? Or do you assume access to a uniform state distribution that you can sample from? The latter is a strong assumption. - The algorithm boxes are generally quite disconnected from the equations in the main text. The paper would be easier to follow if the algorithm box directly referenced equations in the main text (e.g. if Alg 2 line 9 referenced the equation in line 141). - The overall description of the algorithm is terse — the description in the main text is less than a page long. It would be helpful to go into more detail to clearly convey the method. The algorithm boxes are helpful, but including comments in the pseudo code would make it a lot easier to understand, especially given the large amount of non-standard notation. - Minor: The notation for policy entropy is confusing. H_pi(s) is often terminology used to refer to marginal state entropy, rather than policy entropy. Experiments section (more minor comments) - “We set out to demonstrate its effectiveness experimentally” —> If this is actually what the authors did, then this is poor science. The goal of the experiments should be to experimentally test hypotheses and report the results, not to set out to show good results. - Text in the plots is tiny and hard to read